# Unified Segment-to-Segment Framework for Simultaneous Sequence Generation

**Shaolei Zhang**[1,2], **Yang Feng**[1,2*]
[1]Key Laboratory of Intelligent Information Processing,
Institute of Computing Technology, Chinese Academy of Sciences (ICT/CAS)
[2]University of Chinese Academy of Sciences
zhangshaolei20z@ict.ac.cn, fengyang@ict.ac.cn

## Abstract

Simultaneous sequence generation is a pivotal task for real-time scenarios, such as streaming speech recognition, simultaneous machine translation and simultaneous speech translation, where the target sequence is generated while receiving the source sequence. The crux of achieving high-quality generation with low latency lies in identifying the optimal moments for generating, accomplished by learning a mapping between the source and target sequences. However, existing methods often rely on task-specific heuristics for different sequence types, limiting the model's capacity to adaptively learn the source-target mapping and hindering the exploration of multi-task learning for various simultaneous tasks. In this paper, we propose a unified *segment-to-segment framework* (*Seg2Seg*) for simultaneous sequence generation, which learns the mapping in an adaptive and unified manner. During the process of simultaneous generation, the model alternates between waiting for a source segment and generating a target segment, making the segment serve as the natural bridge between the source and target. To accomplish this, Seg2Seg introduces a latent segment as the pivot between source to target and explores all potential source-target mappings via the proposed expectation training, thereby learning the optimal moments for generating. Experiments on multiple simultaneous generation tasks demonstrate that Seg2Seg achieves state-of-the-art performance and exhibits better generality across various tasks[2].

## 1 Introduction

Recently, there has been a growing interest in simultaneous sequence generation tasks [1, 2, 3, 4, 5, 6, 7, 8, 9] due to the rise of real-time scenarios, such as international conferences, live broadcasts and online subtitles. Unlike conventional sequence generation [10], simultaneous sequence generation receives a streaming source sequence and generates the target sequence simultaneously, in order to provide low-latency feedback [11]. To achieve high-quality generation under such low-latency conditions, simultaneous models must learn to establish a mapping between the target sequence and the source sequence [12] and thereby identify the optimal moments for generating [13].

Directly mapping source and target sequences is non-trivial due to inherent differences between the two sequences, such as modalities or languages, resulting in significant representational and structural gaps. For instance, in streaming automatic speech recognition (Streaming ASR) [8, 9, 14, 15, 16], speech needs to be mapped to text, while simultaneous machine translation (SimulMT) [17, 11, 4, 5, 18, 13] requires the mapping from a source language to a target language (i.e., cross-lingual

---

*Corresponding author: Yang Feng

[2]Code is available at: https://github.com/ictnlp/Seg2Seg.

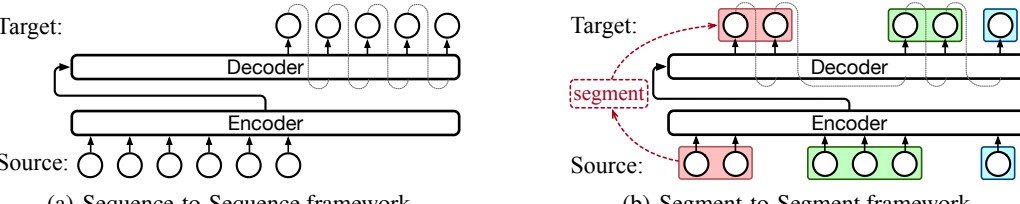

(a) Sequence-to-Sequence framework.  (b) Segment-to-Segment framework.

Figure 1: Illustration of the conventional sequence-to-sequence framework for offline generation and the proposed segment-to-segment framework for simultaneous generation.

alignment [19]). Simultaneous speech translation (SimulST) [1, 2, 3, 20, 21] encounters challenges that encompass both cross-modal and cross-lingual aspects. Therefore, developing an approach to bridge source and target is critical to simultaneous sequence generation.

Existing methods for simultaneous generation often rely on task-specific heuristics to bridge the source and target sequences. For example, streaming ASR methods assume a strong correlation between the target token and local speech, employing a fixed-width window to directly predict the corresponding word [22, 9, 14]. SimulMT methods consider that the source and target sequences have similar lengths, employing fixed wait-k policies [4, 23, 18] or attention mechanisms [5, 24, 25] to establish a token-to-token mapping. Such assumptions of similar length limit their ability to handle sequences with significant length differences [12]. SimulST methods divide the speech into multiple segments to overcome length differences [26, 3], and then apply the fixed wait-k policy [20, 27]. These task-specific heuristics not only hinder the adaptive learning of the source-target mapping but also impede the integration of various simultaneous tasks into a unified framework, restricting the potential of utilizing multi-task learning [28, 29, 30] in simultaneous generation tasks.

Under these grounds, we aim to bridge the source and target sequences in an adaptive and unified manner without any task-specific assumptions. In simultaneous generation process, the model necessarily waits for a source segment and outputs a target segment alternately, with each segment comprising one or more tokens. As such, the source sequence and target sequence should correspond in terms of the segment and ideally agree on the segment representation [31], enabling the segment to serve as a natural bridge between source and target. In this paper, we propose a unified *segment-to-segment framework* (*Seg2Seg*) for simultaneous sequence generation, which introduces latent segments as pivots between source and target. As illustrated in Figure 1, given a streaming source sequence, Seg2Seg determines whether the received source tokens can be aggregated into a latent segment. Once aggregated, the latent segment starts emitting the target tokens until the latent segment can no longer emit any further target tokens. Seg2Seg repeats the above steps until finishing generating. To learn when to aggregate and emit, Seg2Seg employs expectation training to explore all possible source-target mappings and find the optimal moments for generating. Experiments on multiple simultaneous generation tasks, including streaming ASR, SimulMT and SimulST, demonstrate that Seg2Seg achieves state-of-the-art performance and exhibits better generality across various simultaneous tasks.

## 2   Related Work

**Streaming ASR**   Recent streaming ASR methods primarily rely on two approaches: transducer and local attention. Transducer involves a speech encoder and a label predictor, which are aligned via a joiner to determine whether to generate [32, 33, 34]. Local attention approach utilizes monotonic attention to determine whether to generate based on the speech within a local window [6, 7, 35, 22]. Moreover, various methods have been proposed to reduce the latency based on these two approaches by optimizing the alignment process [36, 37, 38, 39, 40].

**SimulMT**   Recent SimulMT methods are mainly based on pre-defined rules or alignment mechanisms. For pre-defined rules, Ma et al. [4] proposed wait-k policy, which waits for $k$ source tokens before alternately waiting/generating one token. Some methods were proposed to improve the flexibility of fixed rules through training [23, 41, 42, 43], simultaneous framework [44, 45], the ensemble of wait-k [46, 18] or post-evaluation [47]. For alignment mechanisms, previous works employ

monotonic attention [5, 24, 31], Gaussian attention [25], binary search [48], non-autoregressive structure [49] or hidden Markov models [13] to learn the alignments between the source and target token-to-token, and make waiting or generating decisions accordingly.

**SimulST** Recent SimulST methods focus on the segmentation of speech [1, 50, 51, 2]. Ma et al. [26] proposed fixed pre-decision to divide speech into equal-length segments, and applied SimulMT methods such as wait-k [4] and MMA [24]. Some other methods first use CTC results [3, 20], ASR results [52] or integrate-and-firing [27] to detect the number of words in speech, and then apply the wait-k policy. Further, Zhang and Feng [53] proposed ITST, which judges whether the received information is sufficient for translation. Zhang et al. [21] proposed MU-ST, which constructs segmentation labels based on meaning units and uses them to train a segmentation model. Zhang and Feng [54] proposed differentiable segmentation (DiSeg) to directly learn segmentation from the underlying translation model via an unsupervised manner.

Previous methods for simultaneous generation often involve task-specific heuristics, which hinder adaptive learning and limit their applicability to other tasks. The proposed Seg2Seg utilizes the latent segments as pivots to achieve fully adaptive learning of source-target mapping. Furthermore, Seg2Seg serves as a unified framework for various simultaneous generation tasks, making multi-task learning in simultaneous generation feasible.

# 3 Method

In this paper, we propose a segment-to-segment framework (Seg2Seg) to map the source sequence to the target sequence, using latent segments as the pivots. With the latent segments, Seg2Seg can adaptively learn to map source to target, enabling it to find the optimal moments to generate the target tokens during the simultaneous generation process. Details are introduced in the following sections.

## 3.1 Mapping with Latent Segments

Seg2Seg leverages the Transformer (encoder-decoder) [55] as the backbone, and further converts the sequence-to-sequence framework to the segment-to-segment framework by introducing latent segments. Formally, we denote the source sequence as $\mathbf{x} = \{x_1, \cdots, x_J\}$ with length $J$, and the target sequence as $\mathbf{y} = \{y_1, \cdots, y_I\}$ with length $I$. In Seg2Seg, the source tokens are first aggregated into several latent segments (source tokens⇒latent segment), and then the latent segment emits the target tokens (latent segment⇒target tokens), as shown in Figure 2.

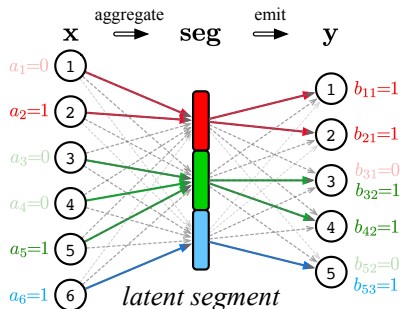

Figure 2: Diagram of source-target mapping with latent segments. The arrows in color and gray represent the mapping in inference and training, respectively.

**Source tokens ⇒ Latent segment** For aggregation, Seg2Seg produces a Bernoulli variable $a_j$ for each source token $x_j$ to determine whether the currently received source tokens can be aggregated into a segment. An aggregation probability $\alpha_j$ is predicted as the parameter for the variable $a_j$, calculated as:

$$\alpha_j = \text{sigmoid}\left(\text{FFN}\left(\text{Rep}\left(x_j\right)\right)\right), \qquad a_j \sim \text{Bernoulli}\left(\alpha_j\right), \tag{1}$$

where FFN $(\cdot)$ is a feed-forward network, Rep $(x_j)$ is the representation of $x_j$, and $\alpha_j$ is aggregation probability at $x_j$. As shown in Figure 2, if $a_j = 0$, Seg2Seg waits for the next input, otherwise, it aggregates the tokens received after the previous segment into a new segment. Once a latent segment is aggregated, we calculate its representation by summing the representations of all the source tokens it contains. Specifically, the representation of the $k^{th}$ latent segment is denoted as $\text{seg}_k$, calculated as:

$$\text{seg}_k = \mathbf{W}^{\text{src}\rightarrow\text{seg}} \sum_{x_j \in \text{seg}_k} \text{Rep}\left(x_j\right), \tag{2}$$

where $\mathbf{W}^{\text{src}\rightarrow\text{seg}}$ is the learnable projection from source to latent segment space.

**Latent segment ⇒ Target tokens** Given latent segment representation, Seg2Seg judges whether $\text{seg}_k$ can emit $y_i$ by producing a Bernoulli variable $b_{ik}$ with the emission probability $\beta_{ik}$ as a

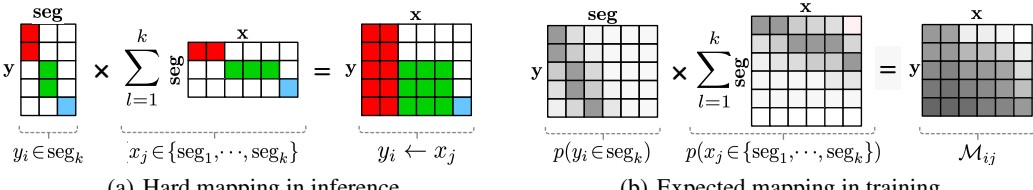

(a) Hard mapping in inference.  (b) Expected mapping in training.

Figure 3: Illustration of source-target mapping in inference and training. (a) The color indicates which latent segment the token belongs to, and the final matrix indicates whether the model receives $x_j$ when generating $y_i$ (i.e., the cross-attention, where the white space is masked out because those source tokens are not received yet.). (b) The shade indicates the probability that the token belongs to different latent segments, and the final matrix indicates the probability that $y_i$ can pay attention to $x_j$.

parameter. Specifically, $b_{ik}$ is calculated as a dot-product form:

$$\beta_{ik} = \text{sigmoid}\left(\frac{\mathbf{W}^{\text{tgt}\rightarrow\text{seg}}\text{Rep}\,(y_{i-1}) \cdot \text{seg}_k^\top}{\sqrt{d}}\right), \qquad b_{ik} \sim \text{Bernoulli}\,(\beta_{ik}), \qquad (3)$$

where $\mathbf{W}^{\text{tgt}\rightarrow\text{seg}}$ is the learnable projection from target to latent segment space, and $d$ is the input dimension. During emission, if $b_{ik} = 1$, Seg2Seg generates $y_i$ based on the current received source tokens, otherwise it stops emitting and waits for the next input. Take Figure 2 as an example, $y_3$ will not be emitted from $\text{seg}_1$ as $b_{31} = 0$. After aggregating $\text{seg}_2$, $y_3$ is emitted from $\text{seg}_2$ as $b_{32} = 1$.

Overall, Seg2Seg alternates between waiting for enough source tokens to aggregate a latent segment (i.e., wait until $a_j = 1$), and outputting the target tokens until the current latent segment can no longer emit any further tokens (i.e., output until $b_{ik} = 0$). Take the mapping in Figure 2 for instance, Seg2Seg waits for 2 source tokens and generates 2 target tokens, then waits for 3 and generates 2 tokens, then waits for 1 and generates 1 token. Figure 3(a) gives the corresponding mapping in matrix form, where the final matrix indicates whether the model receives $x_j$ when generating $y_i$.

## 3.2  Training

During training, Seg2Seg tends to learn the aggregation and emission in an adaptive manner. However, a significant challenge arises from the use of Bernoulli variables $a_j$ and $b_{ik}$ for aggregation and emission, which prevents the back-propagation [56, 57] to the aggregation probability $\alpha_j$ and emission probability $\beta_{ik}$. To address this issue, we propose expectation training that employs $\alpha_j$ and $\beta_{ik}$ instead of Bernoulli variables to calculate the expected mapping, which can be jointly trained with the underlying Transformer model. As illustrated in Figure 3, in expectation training, the source tokens and target tokens are no longer forced to be associated with a single latent segment, but rather can belong to multiple latent segments by probability.

For the aggregation process from source tokens to latent segment, we introduce $p\,(x_j \in \text{seg}_k)$ to represent the probability that $x_j$ belongs to the latent segment $\text{seg}_k$. Since the aggregation process is monotonic with the streaming source sequence, i.e., which segment $x_j$ belongs to is only related to $x_{j-1}$, $p\,(x_j \in \text{seg}_k)$ can be calculated via dynamic programming:

$$p\,(x_j \in \text{seg}_k) = p\,(x_{j-1} \in \text{seg}_{k-1}) \times \alpha_{j-1} + p\,(x_{j-1} \in \text{seg}_k) \times (1 - \alpha_{j-1}). \qquad (4)$$

We consider all possible latent segments in the expectation training, so $k$ ranges from 1 to $J$ (i.e., aggregate at most $J$ segments with one token in each segment), even if the source tokens may belong to the later latent segment with a small probability, as shown in Figure 3(b). With $p\,(x_j \in \text{seg}_k)$, we calculate the expected representation of latent segment by weighting all source tokens:

$$\text{seg}_k = \mathbf{W}^{\text{src}\rightarrow\text{seg}} \sum_{j=1}^{J} p\,(x_j \in \text{seg}_k) \times \text{Rep}\,(x_j). \qquad (5)$$

For the emission process from latent segment to target tokens, we introduce $p\,(y_i \in \text{seg}_k)$ to represent the probability that $y_i$ can be emitted from latent segment $\text{seg}_k$. Since the emission process is mono-

tonic with the simultaneous generation, $p\left(y_i \in \text{seg}_k\right)$ can be calculated via dynamic programming:

$$p\left(y_i \in \text{seg}_k\right) = \beta_{i,k} \sum_{l=1}^{k} \left( p\left(y_{i-1} \in \text{seg}_l\right) \prod_{m=l}^{k-1} \left(1 - \beta_{i,m}\right) \right). \tag{6}$$

We give a detailed introduction to the dynamic programming algorithm in Appendix A.

**Learning Mapping**  To adaptively learn $\boldsymbol{\alpha}$ and $\boldsymbol{\beta}$, we jointly train $p\left(x_j \in \text{seg}_k\right)$ and $p\left(y_i \in \text{seg}_k\right)$ with Transformer via the cross-entropy loss $\mathcal{L}_{ce}$. During inference, each target token in Seg2Seg no longer focuses on all source tokens, but can only pay attention to the source token within the same latent segment or the previous segments (i.e., the current received tokens), as shown in Figure 3(a). So in training, we calculate the probability that $y_i$ can pay attention to $x_j$, denoted as $\mathcal{M}_{ij}$:

$$\mathcal{M}_{ij} = \sum_{k} p\left(y_i \in \text{seg}_k\right) \times p\left(x_j \in \{\text{seg}_1, \cdots, \text{seg}_k\}\right) = \sum_{k} p\left(y_i \in \text{seg}_k\right) \times \sum_{l=1}^{k} p\left(x_j \in \text{seg}_l\right). \tag{7}$$

Then, we multiply the mapping $\mathcal{M}_{ij}$ with the original cross-attention [53, 58] and normalize it to get the final attention distribution, which is used to calculate the expected target representation. By jointly training mapping and generation via the cross-entropy loss $\mathcal{L}_{ce}$, Seg2Seg will assign higher $\mathcal{M}_{ij}$ between those related source and target tokens, thereby learning a reasonable mapping.

**Learning Latency**  Besides learning mapping for high-quality generation, we also introduce a latency loss $\mathcal{L}_{latency}$ to encourage low latency. We utilize two commonly used latency metrics, consecutive wait (CW) [17] and average lagging (AL) [4], to calculate the expected latency of Seg2Seg, where CW measures the number of latent segments (i.e., streaming degree [25]), and AL measures the lagging of target token (i.e., lagging degree). Therefore, $\mathcal{L}_{latency}$ is calculated as:

$$\mathcal{L}_{latency} = \mathcal{C}_{\text{CW}}\left(\boldsymbol{\alpha}\right) + \mathcal{C}_{\text{AL}}\left(\boldsymbol{\mathcal{M}}\right),$$

$$\text{where} \quad \mathcal{C}_{\text{CW}}\left(\boldsymbol{\alpha}\right) = \left\| \sum_{j=1}^{|\mathbf{x}|} \alpha_j - \lambda\left|\mathbf{y}\right| \right\|_2 + \left\| \sum \text{MaxPool}\left(\alpha_i, \left\lfloor \frac{|\mathbf{x}|}{\lambda\left|\mathbf{y}\right|} \right\rfloor \right) - \lambda\left|\mathbf{y}\right| \right\|_2, \tag{8}$$

$$\mathcal{C}_{\text{AL}}\left(\boldsymbol{\mathcal{M}}\right) = \frac{1}{|\mathbf{y}|} \sum_{i=1}^{|\mathbf{y}|} \sum_{j=1}^{|\mathbf{x}|} \mathcal{M}_{ij}.$$

For the number of latent segments $\mathcal{C}_{\text{CW}}\left(\boldsymbol{\alpha}\right)$, following Zhang and Feng [54], we constrain Seg2Seg via the expected segment number $\sum_{j=1}^{|\mathbf{y}|} \alpha_j$ and the uniformity of aggregation, where $\text{MaxPool}\left(\cdot\right)$ is the max polling operation with kernel size of $\left\lfloor \frac{|\mathbf{x}|}{\lambda|\mathbf{y}|} \right\rfloor$. For the expected lagging $\mathcal{C}_{\text{AL}}\left(\boldsymbol{\mathcal{M}}\right)$, we constrain the expected lagging $\sum_{j=1}^{|\mathbf{y}|} \mathcal{M}_{ij}$ of target token $y_i$. $\lambda$ is a hyperparameter that controls the overall latency of Seg2Seg. A larger $\lambda$ encourages Seg2Seg to aggregate more latent segments, thereby achieving low latency. When $\lambda \to 0$, the number of latent segments decreases and latency becomes higher, finally degenerating into a sequence-to-sequence framework when $\lambda = 0$.

Overall, the total training objective of Seg2Seg is the trade-off between $\mathcal{L}_{ce}$ for generation quality and $\mathcal{L}_{latency}$ for generation latency, calculated as:

$$\mathcal{L} = \mathcal{L}_{ce} + \mathcal{L}_{latency}. \tag{9}$$

### 3.3  Inference

In inference, we set $a_j = 1$ when $\alpha_j \geq 0.5$ and $b_{ik} = 1$ when $\beta_{ik} \geq 0.5$ without sampling [24, 13]. Algorithm 1 illustrates the specific inference process of Seg2Seg. Given a streaming source sequence, Seg2Seg continuously repeats the process of aggregating source tokens into a latent segment (lines 2-6) when $a_j = 1$ and then emitting target tokens from the latent segment (lines 8-12) while $b_{ik} = 1$, until the generation process is completed.

Owing to generating the target sequence in units of segment, it is natural for Seg2Seg to use beam search inside each target segment. Therefore, in the following experiments, we set the size of the beam search for each segment to 5.

**Algorithm 1** Inference of Segment-to-Segment Framework

---

**Input:** Source sequence $\mathbf{x}$.
**Output:** Target sequence $\hat{\mathbf{y}}$.
**Initialization:** Received source sequence $\hat{\mathbf{x}} = [\,]$; Target sequence $\hat{\mathbf{y}} = [\langle\text{bos}\rangle]$; Index $j = 0$, $i = 1$, $k = 1$.
 1: **while** $\hat{y}_{i-1} \neq \langle\text{eos}\rangle$ **do**
 2:
 3:    **while** $a_j == 0$ and $\hat{\mathbf{x}} \neq \mathbf{x}$ **do**                                    ▷ WAIT
 4:        Wait for the next source token $x_{j+1}$;
 5:        $\hat{\mathbf{x}} \leftarrow \hat{\mathbf{x}} + x_{j+1}$;
 6:        $j \leftarrow j + 1$;
 7:    **end**
 8:
 9:    Calculate representation of latent segment $\text{seg}_k$ according to Eq.(2);
10:
11:    **while** ($b_{ik} == 1$ or $\hat{\mathbf{x}} == \mathbf{x}$) and $\hat{y}_{i-1} \neq \langle\text{eos}\rangle$ **do**          ▷ Generate
12:        Generate the target token $\hat{y}_i$ based on $\hat{\mathbf{x}}$;
13:        $\hat{\mathbf{y}} \leftarrow \hat{\mathbf{y}} + \hat{y}_i$;
14:        $i \leftarrow i + 1$;
15:    **end**
16:
17:    $k \leftarrow k + 1$;
18: **end**
19: **return** $\hat{\mathbf{y}}$;

---

# 4 Experiments

## 4.1 Datasets

We conduct experiments on the most common benchmarks of three representative simultaneous generation tasks, including streaming ASR, SimulMT and SimulST.

**Streaming ASR**  We apply LibriSpeech[3] benchmark [59], which consists of 960 hours English audio. We use `dev-clean` (5.4 hours) and `dev-other` (5.3 hours) as validation sets, and `test-clean` (5.4 hours) and `test-other` (5.1 hours) as test sets, where `test-other` set contains more noisy audio. For speech, we use the raw 16-bit 16kHz mono-channel audio wave. For text, we use SentencePiece [60] to generate a unigram vocabulary of size 10000.

**SimulMT**  We apply WMT15[4] German→English (De→En) benchmark, including 4.5M sentence pairs for training. We use `newstest2013` as validation set (3000 pairs), and `newstest2015` as test set (2169 pairs). 32K BPE [61] is applied and vocabulary is shared across languages.

**SimulST**  We apply MuST-C[5] English→German (En→De) (408 hours, 234K pairs) and English → Spanish (En→Es) (504 hours, 270K pairs) benchmarks [62]. We use `dev` as validation set (1423 pairs for En→De, 1316 pairs for En→Es) and `tst-COMMON` as test set (2641 pairs for En→De, 2502 pairs for En→Es), respectively. The pre-processing is the same as streaming ASR tasks.

## 4.2 Systems Settings

We conducted experiments on several strong baselines for all three tasks, described as follows.

**Offline** [55] model waits for the complete source sequence before generating the target sequence. Offline model is decoded with beam 5.

**# Streaming Automatic Speech Recognition (Streaming ASR)**

**T-T** [34] uses Transformer Transducer to determine waiting/generating via alignments from the joiner between the speech encoder and text predictor. Some methods, including ConstAlign [36], FastEmit [37] and SelfAlign [38] are proposed to further reduce the latency of the Transducer.

---

[3] https://www.openslr.org/12
[4] https://www.statmt.org/wmt15/
[5] https://ict.fbk.eu/must-c

**MoChA** [22] applies monotonic chunkwise attention to generate the target token based on the speech within a local window. Various training methods, such as DeCoT [39], MinLT [39] and CTC [40], are proposed to further constrain the latency of MoChA.

# Simultaneous Machine Translation (SimulMT)

**Wait-k** [4] first waits for $k$ source tokens, and then alternately generates and waits for one token.

**Multipath Wait-k** [23] trains a wait-k model via randomly sampling different $k$ between batches.

**MoE Wait-k** [18] applies mixture-of-experts (MoE) to learn multiple wait-k policies during training.

**Adaptive Wait-k** [46] trains a set of wait-k models (e.g., from wait-1 to wait-13), and heuristically composites these models based on their outputs during inference.

**MMA** [24] applies monotonic multi-head attention and predicts a variable to indicate waiting or generating, which are trained through monotonic attention [35].

**GMA** [25] introduces Gaussian multi-head attention and uses a Gaussian prior to learn the alignments via attention. With alignments, GMA decides when to start translating based on the aligned positions.

**GSiMT** [63] generates waiting/generating decisions, and considers all possible decisions in training.

**HMT** [63] proposes Hidden Markov Transformer, which uses a HMM to correspond translating moments with the target tokens, thereby learning the optimal translating moments for generating.

# Simultaneous Speech Translation (SimulST)

**Wait-k, MMA** [26] for SimulMT can be applied to SimulST by making a fixed pre-decision to split the speech into $280ms$ durations, where each duration corresponds to one word.

**Wait-k-Stride-n** [20] generates $n$ tokens every $n{\times}280ms$ to address the issue of length differences between speech and text. We set $n{=}2$ following their best result.

**SimulSpeech** [3] divides the speech based on a CTC word detector, and then applies wait-k policy.

**SH** [52] uses the shortest hypothesis in ASR results as word number, and then applies wait-k policy.

**RealTrans** [20] detects the word number in the streaming speech via counting the blank in CTC results, and then applies the wait-k-stride-n policy.

**MMA-CMDR** [64] incorporates cross-modal decision regularization to MMA, which leverages the transcription of speech to improve the decision of MMA.

**MoSST** [27] uses the integrate-and-firing method [65] to segment the speech based on the cumulative acoustic information, and then applies the wait-k policy.

**ITST** [53] quantifies the transported information from source to target, and subsequently determines whether to generate output based on the accumulated received information.

**MU-ST** [21] trains an external segmentation model based on the constructed data to detect the meaning unit, and uses it to decide whether to generate.

**DiSeg** [54] jointly learns the speech segmentation with the underlying translation model via the proposed differentiable segmentation in an unsupervised manner.

All implementations are adapted from Fairseq Library [66]. In Seg2Seg, we use the standard Transformer-Base (6 encoder and 6 decoder layers) [55] for SimulMT. For streaming ASR and SimulST, we replace the word embedding layer in Transformer-Base with a pre-trained Wav2Vec2.0[6] [67] to extract the acoustic embedding, and the rest remains the same as SimulMT.

**Evaluation**   We use `SimulEval` [7] [68] to evaluate the quality and latency of simultaneous generation. For streaming ASR, following Inaguma and Kawahara [40], we use word error rate (WER) for the quality and the mean alignment delay for the latency, which considers the average difference between generating moments and ground-truth alignments. For SimulMT, following Ma et al. [4] and Zhang and Feng [13], we use BLEU [69] for the generation quality and average lagging (AL) [4] for latency, which measures the average offsets that outputs lag behind inputs (using token as the unit of AL). For SimulST, following Ma et al. [26], we use sacreBLEU [70] for the generation quality and average lagging (AL) for latency (using millisecond $ms$ as the unit of AL). Refer to Appendix C for the detailed calculation of the latency metrics.

---

[6]`dl.fbaipublicfiles.com/fairseq/wav2vec/wav2vec_small.pt`
[7]`https://github.com/facebookresearch/SimulEval`

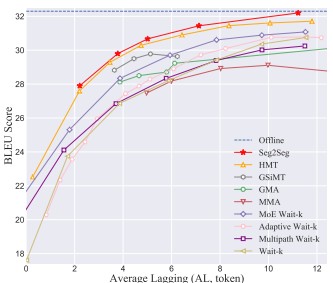

Figure 4: SimulMT results of quality v.s. latency (AL, tokens) on WMT15 De→En.

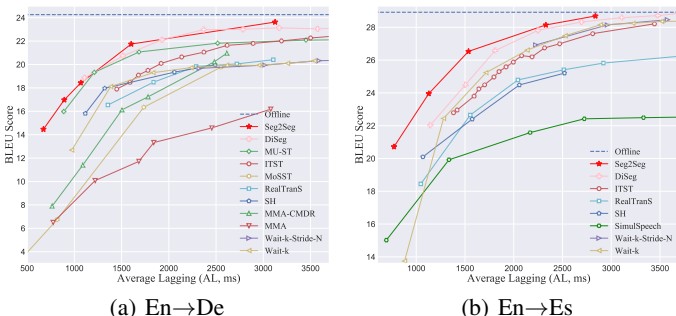

(a) En→De        (b) En→Es

Figure 5: SimulST results of quality v.s. latency (AL, $ms$) on MuST-C En→De and En→Es.

## 4.3 Main Results

**Results on Streaming ASR**   Table 1 reports the results on streaming ASR, where Seg2Seg exhibits a better trade-off between generation quality and latency. Compared with T-T using the transducer to align speech and text token-to-token [34], or MoChA generating text based on local speech [22], Seg2Seg adaptively learns source-target mapping through latent segments, thereby performing better.

**Results on SimulMT**   Consistent with the previous methods [4, 24, 13], we adjust the value of $\lambda$ (refer to Eq.(8)) to show the performance of Seg2Seg under varying latency. Figure 4 shows that Seg2Seg outperforms previous SimulMT methods at all latency. Compared to methods based on pre-defined rules, such as wait-k and

Table 1: Streaming ASR results.

| Systems | WER(↓) clean | WER(↓) other | Latency(↓) |
|---|---|---|---|
| Offline | 3.51 | 8.49 | - |
| T-T | 3.40 | 9.50 | 610 |
|   +ConstAlign | 4.00 | 11.10 | 328 |
|   +FastEmit | 4.00 | 10.40 | 195 |
|   +SelfAlign | 4.00 | 10.70 | 145 |
| MoChA | 4.80 | 14.20 | 320 |
|   +CTC | 4.00 | 11.20 | 240 |
|   +DeCoT | 3.90 | 11.60 | 240 |
|   +MinLT | 4.50 | 11.70 | 320 |
| Seg2Seg | 3.55 | 8.73 | 324 |

MoE wait-k, Seg2Seg is more flexible in making generating/waiting decisions, achieving significant advantages. Other methods, such as MMA, GMA and HMT, align the target and source token-to-token. However, since the alignment between the two languages may not be one-to-one [4], some local reordering and multi-word structures can affect the performance [12]. By mapping source to target at the segment level, Seg2Seg is more in line with the simultaneous generation process and mitigates these issues, ultimately achieving state-of-the-art performance.

**Results on SimulST**   For the most challenging SimulST in Figure 5, Seg2Seg achieves state-of-the-art performance, especially at low latency. Most of the previous SimulST methods either segment the speech into fixed lengths [26, 64] or detect the number of words in the speech [3, 52, 20, 27] and then apply a wait-k policy, where both non-derivable word detection and wait-k policy hinder the adaptive learning of the model. Owing to the proposed expectation training, Seg2Seg is completely differentiable and able to jointly learn the mapping from the source to the latent segment and from the latent segment to the target, thereby finding the optimal moments for generating.

## 4.4 Superiority of Unified Framework on Multi-task Learning

In the sequence-to-sequence framework, multi-task learning composed of ASR, MT and ST is shown to improve the performance on difficult tasks (e.g. speech translation) by sharing knowledge among different tasks [71, 72, 73]. However, in previous simultaneous generation methods, different tasks often involve different architectures and heuristics, leaving no room for multi-task learning. Owing to not involving any task-related heuristics, the proposed unified segment-to-segment framework provides a possibility to apply multi-task learning in simultaneous generation. In Seg2Seg, multi-task learning can include streaming ASR, SimulMT and SimulST, and these three tasks share all parameters, except that SimulMT has a text embedding and streaming ASR/SimulST have a shared speech embedding.

Figure 6 demonstrates the improvements brought by multi-task learning on the most challenging SimulST task. By employing multi-task learning in a unified framework, Seg2Seg can achieve further improvements. Specifically, jointly training with streaming ASR yields more discernible improvements, which is mainly because the monotonic properties between speech and text inherent in streaming ASR assist SimulST in learning the source-target mapping [52, 27, 64, 54]. Therefore, the unified Seg2Seg facilitates the sharing of knowledge among various simultaneous tasks through multi-task learning and is helpful for the difficult tasks, such as SimulST.

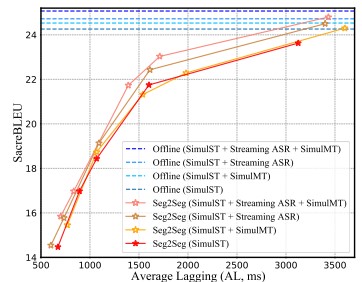

Figure 6: SimulST results on MuST-C En→De with multi-task learning.

## 5 Analysis

We conducted extensive analyses to investigate the specific improvements of Seg2Seg. Unless otherwise specified, all the results are reported on SimulST with MuST-C En→De test set, which is more difficult simultaneous generation task. Refer to Appendix B for more extended analyses.

### 5.1 Improvements of Adaptive Learning

Seg2Seg learns the mappings from source to segment and from segment to target in an adaptive manner, without any task-specific assumptions. To verify the effect of adaptive learning, we respectively replace the source-to-segment and segment-to-target mappings with heuristic rules, such as fixed-length segment (i.e., fixed-seg) [26] and wait-k/wait-k-stride-n policy (i.e., fixed-emit) [4, 20], and show the SimulST En→De results in Figure 7.

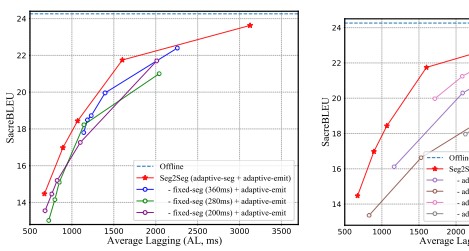

(a) w/o adaptive aggregation.     (b) w/o adaptive emission.

Figure 7: Improvements brought by adaptive learning.

The results show that adaptive learning significantly outperforms heuristic rules. Compared with dividing the source into fixed lengths of $200/280/360ms$, Seg2Seg can adaptively determine whether the received source token can be aggregated into a latent segment, bringing about 1 BLEU improvement. Compared with the rule-based wait-k policy, Seg2Seg judges whether to emit the target token based on the latent segment, thus finding more reasonable generating moments [5].

### 5.2 Quality of Aggregation and Emission

Seg2Seg learns aggregation and emission adaptively, so we further explore the quality of aggregation and emission, respectively. We apply streaming ASR and SimulMT tasks for evaluation. The detailed calculation of the metrics for aggregation and emission quality are shown in Appendix B.1.

**Aggregation Quality** To verify whether Seg2Seg can aggregate the source tokens into a latent segment at the appropriate moments in streaming ASR, following Zhang and Feng [54], we conduct experiments on the Buckeye dataset[8] [78], which is a speech segmentation benchmark with the annotated word boundaries. Table 2 shows the segmentation quality of Seg2Seg with some segmentation baselines, and the metrics include precision (P), recall (R) and R-value (comprehensive score) [79]. Seg2Seg achieves better segmentation precision and higher

Table 2: Segmentation accuracy of Seg2Seg.

| Systems | P(↑) | R(↑) | R-value(↑) |
|---|---|---|---|
| ES K-Means [74] | 30.7 | 18.0 | 39.7 |
| BES GMM [75] | 31.7 | 13.8 | 37.9 |
| VQ-CPC [76] | 18.2 | 54.1 | -86.5 |
| VQ-VAE [76] | 16.4 | 56.8 | -126.5 |
| DSegKNN [77] | 30.9 | 32.0 | 40.7 |
| Fixed($280ms$) | 28.1 | 16.3 | 38.4 |
| Seg2Seg | 41.1 | 18.1 | **41.2** |

[8]https://buckeyecorpus.osu.edu

comprehensive score R-value, showing that Seg2Seg can perform aggregation and segment the speech at reasonable moments (i.e., token boundaries instead of breaking the continuous speech of a word [27]).

**Emission Quality**  To verify whether the model emits at reasonable moments, we follow Zhang and Feng [31] to evaluate the emission quality in SimulMT based on alignments. We apply RWTH[9] De→En alignment dataset, and calculated the proportion of the model emitting the target token after receiving its aligned source token, used as the emission accuracy. Figure 8 shows that Seg2Seg can receive more aligned source tokens before emitting under the same latency, meaning that Seg2Seg finds more favorable moments for generating and achieves high-quality generation.

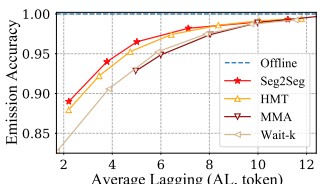

Figure 8: Emission accuracy of Seg2Seg in SimulMT.

### 5.3 Effect of Latent Segments

To investigate whether the latent segments can act as a bridge between the source and target, we calculate the representations of the source tokens ($\sum_{x_j \in seg_k} x_j$), target tokens ($\sum_{y_i \in seg_k} y_i$), and latent segment $seg_k$ within each segment during SimulST on En→De. We then apply the T-SNE dimensionality reduction algorithm to project these representations into a 2D space. By doing so, we obtain a bivariate kernel density estimation of the representation distribution of source segment, target segment and latent segments, depicted in Figure 9. The visualization clearly demonstrates that the latent segment locates between the source and target sequences in the representation space, effectively serving as a bridge connecting the source and target.

Furthermore, we calculate the cosine similarity between the representations of the source, target and latent segments, as shown in Table 3. It is evident that the similarity between the source and target representations is low, posing a challenge for the model to directly map the source to the target. Conversely, the similarity between the latent segment and the source, as well as the latent segment and the target, is significantly higher. Hence, by introducing the latent segment as a pivot, the model can more easily learn the mapping from the source to the latent segment, and subsequently from the latent segment to the target, thereby finding the optimal moments for generating and achieving better performance.

Table 3: Representational similarity with the latent segment.

|  | Similarity |
| --- | --- |
| source ⇔ target | 0.53 % |
| source ⇔ segment | 20.01 % |
| segment ⇔ target | 14.66 % |

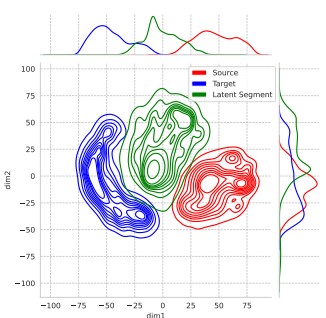

Figure 9: Bivariate kernel density estimation visualization on the representations of source, target and latent segment.

## 6 Conclusion

In this paper, we propose a unified segment-to-segment framework for simultaneous sequence generation, which bridges the source and target sequences using latent segments as pivots. Unified Seg2Seg enables the handling of multiple simultaneous generation tasks and facilitates multi-task learning. Experiments and analyses show the superiority of Seg2Seg on performance and generalization.

## Limitations

The proposed Seg2Seg employs the encoder-decoder architecture as its backbone, and exhibits better generality across multiple simultaneous generation tasks. In addition to its primary application on generation tasks, the encoder (aggregation process) or decoder (emission process) of Seg2Seg can also be separately used for some real-time tasks based on encoder-only or decoder-only architecture, such as streaming tagging and online parsing. We leave this for further exploration in future work.

---

[9] https://www-i6.informatik.rwth-aachen.de/goldAlignment/

## Acknowledgements

We thank all the anonymous reviewers for their insightful and valuable comments.

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

# A  Dynamic Programming of Mapping

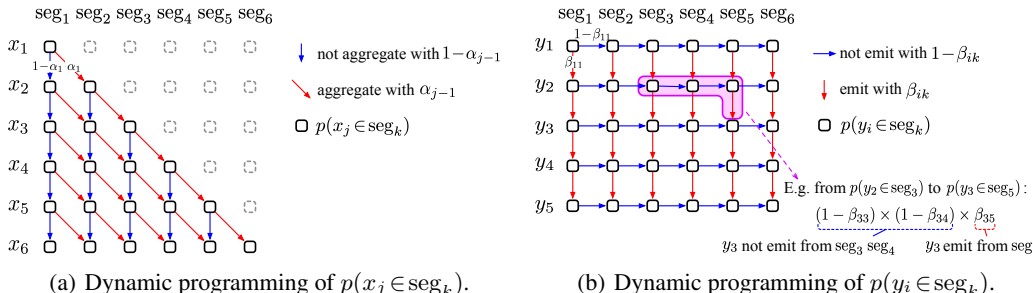

Figure 10: Schematic diagram of the dynamic programming of mapping in expectation training.

In Sec.3.2, we propose expectation training for Seg2Seg to learn when to aggregate and emit through $p\left(x_j \in \text{seg}_k\right)$ and $p\left(y_i \in \text{seg}_k\right)$. Given the monotonic property of simultaneous sequence generation tasks, we can calculate $p\left(x_j \in \text{seg}_k\right)$ and $p\left(y_i \in \text{seg}_k\right)$ using a dynamic programming algorithm. In the following sections, we will provide a detailed explanation of the dynamic programming approach.

## A.1  Source Tokens to Latent Segment

Given the streaming source sequence, Seg2Seg predicts aggregation probability $\alpha_j$ for $x_j$ to represent the probability of aggregating the received source tokens into a latent segment at $x_j$. Given aggregation probability $\alpha_j$, we calculate $p\left(x_i \in \text{seg}_k\right)$ via dynamic programming.

The whole aggregation process is monotonic with inputs, which means that the $(k+1)^{th}$ latent segment can only be aggregated once the $k^{th}$ segment has already been aggregated. Additionally, each latent segment must be aggregated by at least one source token, otherwise a latent segment without representation would be meaningless. As a result, whether $x_j$ belongs to latent segment $\text{seg}_k$ depends on which segment that $x_{j-1}$ is located in, consisting of 3 situations:

- If $x_{j-1} \in \text{seg}_{k-1}$:  As illustrated by the red line in Figure 10(a), $x_j$ belongs to latent segment $\text{seg}_k$ when Seg2Seg aggregate at $x_{j-1}$ with probability $\alpha_{j-1}$;

- If $x_{j-1} \in \text{seg}_k$:  As illustrated by the blue line in Figure 10(a), $x_j$ belongs to latent segment $\text{seg}_k$ (i.e., the same latent segment with $x_{j-1}$) when Seg2Seg does not aggregate at $x_{j-1}$ with probability $1 - \alpha_{j-1}$;

- Otherwise:  $x_j$ can not belong to $\text{seg}_k$ anyway, i.e., with probability 0.

By combining these situations, $p\left(x_i \in \text{seg}_k\right)$ is calculated as:

$$p\left(x_j \in \text{seg}_k\right) = p\left(x_{j-1} \in \text{seg}_{k-1}\right) \times \alpha_{j-1} + p\left(x_{j-1} \in \text{seg}_k\right) \times \left(1 - \alpha_{j-1}\right), \qquad (10)$$

where the initialization is

$$p(x_1 \in \text{seg}_k) = \begin{cases} 1 & \text{if } k = 1 \\ 0 & \text{if } k \neq 1 \end{cases}, \qquad (11)$$

because the first source token inevitably belongs to the first segment. With the above dynamic programming algorithm, we can calculate $p\left(x_j \in \text{seg}_k\right)$, for $j = 1, \cdots, J$ and $k = 1, \cdots, J$.

## A.2  Latent Segment to Target Tokens

After getting the latent segments, Seg2Seg predicts the emission probability $\beta_{ik}$, which indicates the probability of emitting the target token $y_i$ from the latent segment $\text{seg}k$. With the emission probability $\beta ik$, we compute $p\left(y_i \in \text{seg}_k\right)$ using dynamic programming as well. Note that there is one difference between the aggregation process and the emission process when employing dynamic programming. In the aggregation process, each latent segment must be aggregated by at least one

source token. However, in the emission process, the latent segment has the option to not generate any target token, as not all source tokens have corresponding target tokens.

Whether $y_i$ belongs to latent segment $\text{seg}_k$ depends on which segment that $y_{i-1}$ is emitted from, consisting of 3 situations:

- If $y_{i-1} \in \text{seg}_k$: $y_i$ is emitted from latent segment $\text{seg}_k$ with probability $\beta_{ik}$;
- If $y_{i-1} \in \text{seg}_l$ for $l = 1, \cdots, k-1$: $y_i$ is emitted from latent segment $\text{seg}_k$ when $y_i$ is not emitted from $\text{seg}_l$ to $\text{seg}_{k-1}$, and then emitted from $\text{seg}_k$. Taking Figure 10(b) as an example, if $y_2 \in \text{seg}_3$, the premise of $y_3 \in \text{seg}_5$ is that $y_3$ is not emitted from $\text{seg}_3$ and $\text{seg}_4$, and is emitted from $\text{seg}_5$. Formally, the probability is calculated as:

$$\beta_{ik} \times \prod_{m=l}^{k-1} (1 - \beta_{i,m}), \tag{12}$$

  where $\prod_{m=l}^{k-1} (1 - \beta_{i,m})$ is the probability that $y_i$ is not emitted from $\text{seg}_l$ to $\text{seg}_{k-1}$;
- If $y_{i-1} \in \text{seg}_l$ for $l = k+1, \cdots$: $y_i$ can not be emitted from $\text{seg}_k$ anyway, as the emission process is monotonic, i.e., with probability 0.

By combining these situations, $p\left(y_i \in \text{seg}_k\right)$ is calculated as:

$$p\left(y_i \in \text{seg}_k\right) = \beta_{i,k} \sum_{l=1}^{k} \left( p\left(y_{i-1} \in \text{seg}_l\right) \prod_{m=l}^{k-1} (1 - \beta_{i,m}) \right), \tag{13}$$

where the initialization is

$$p\left(y_1 \in \text{seg}_k\right) = \beta_{1,k} \prod_{m=1}^{k-1} (1 - \beta_{1,m}). \tag{14}$$

With the above dynamic programming algorithm, we can calculate $p\left(y_i \in \text{seg}_k\right)$, for $i = 1, \cdots, I$ and $k = 1, \cdots, J$.

# B  Extended Analyses

## B.1  Detailed Calculation of Aggregation and Emission Quality

In Sec.5.2, we evaluate the aggregation and emission quality of Seg2Seg. Here, we give detailed calculations of aggregation and emission quality.

**Aggregation Quality**   We verify whether Seg2Seg can aggregate (segment) the source speech sequence at the appropriate moments on the speech segmentation task [80]. For our evaluation, we utilize the Buckeye dataset, where the ground-truth segmentation is based on word units. Evaluation metrics consist of precision (P), recall (R), and the comprehensive score R-value. Note that since the ground-truth segmentation in Buckeye is in units of words, and the aggregation of Seg2Seg is in units of segments, which makes the segment number of Seg2Seg may be less than the ground-truth segment number, so precision can better reflect whether the aggregation moments of Seg2Seg is reasonable. R-value [79] is a more robust comprehensive metric for speech segmentation task, calculated as:

$$\text{R-value} = 1 - \frac{|r_1| + |r_2|}{2}, \tag{15}$$

$$\text{where} \quad r_1 = \sqrt{(1-R)^2 + \left(\frac{R}{P} - 1\right)^2}, \qquad r_2 = \frac{-\left(\frac{R}{P} - 1\right) + R - 1}{\sqrt{2}}. \tag{16}$$

A larger R-value indicates better segmentation quality, where R-value $= 100\%$ if and only if $P = 100\%$ and $R = 100\%$.

**Emission Quality**   We verify whether Seg2Seg can emit the target token at appropriate moments in SimulMT. In simultaneous machine translation, it is crucial for the model to emit the corresponding target token after receiving its aligned source token [5], so the alignments can be used as the basis for

judging whether the emitting moments are reasonable. Following Zhang and Feng [31],Guo et al. [47], Zhang and Feng [13], we calculate the proportion of the ground-truth aligned source tokens received before emitting as the emission quality. We apply RWTH[10] De→En alignment dataset and denote the ground-truth aligned source position of $y_i$ as $a_i$, while use $t_i$ to record the emitting moments of $y_i$. Then, the emission quality is calculated as:

$$\text{Score} = \frac{1}{|\mathbf{y}|} \sum_{i=1}^{|\mathbf{y}|} \mathbb{1}_{a_i \leq t_i}, \quad \text{where} \quad \mathbb{1}_{a_i \leq t_i} = \begin{cases} 1, & a_i \leq t_i \\ 0, & a_i > t_i \end{cases}. \tag{17}$$

## B.2 Visualization of Mapping

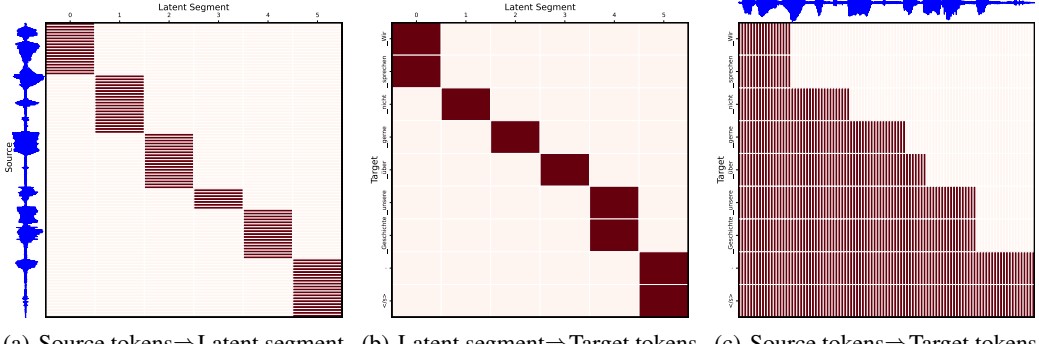

(a) Source tokens⇒Latent segment  (b) Latent segment⇒Target tokens  (c) Source tokens⇒Target tokens

Figure 11: Visualization of the mapping with the latent segment during inference on En→De SimulST (Case #2258). The red rectangles in (a) and (b) indicate the correspondence between token and latent segment. (c) indicates the received source tokens when generating each target token. The English meaning of target sequence: Wir (_we) sprechen (_speak) nicht (_not) gerne (_gladly) über (_about) unsere (_our) Geschichte (_story).

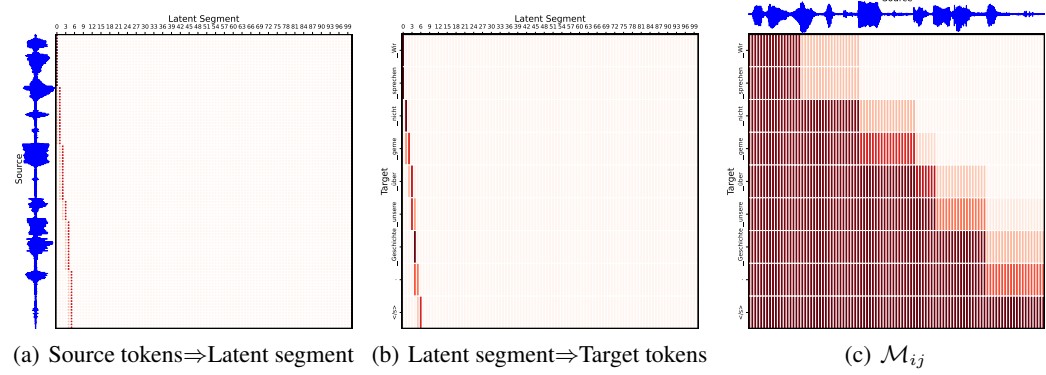

(a) Source tokens⇒Latent segment  (b) Latent segment⇒Target tokens  (c) $\mathcal{M}_{ij}$

Figure 12: Visualization of the mapping with the latent segment during training on En→De SimulST (Case #2258). The shade of red rectangles in (a) and (b) indicate the probability that the token belongs to different latent segments. (c) indicates the probability that $y_i$ can pay attention to $x_j$. Note that Seg2Seg considers all possible latent segments during training, so there are more latent segments in (a) and (b) than inference.

We visualize the proposed source-target mapping with the latent segment as the pivot in Figure 11 and 12, where the case is from the most challenging SimulST task.

**Inference** Figure 11 shows the mapping during inference. Seg2Seg effectively aggregates a lengthy speech sequence (approximately 101 tokens) into 6 latent segments, and then these 6 latent segments

---

[10]https://www-i6.informatik.rwth-aachen.de/goldAlignment/

emit 8 target tokens. As depicted in Figure 11(a), Seg2Seg exhibits high-quality aggregation by selectively splitting and aggregating the speech at appropriate boundaries, thereby preserving the integrity of the acoustic information [27]. During emission, Seg2Seg exhibits the ability to generate multi-word structures, such as '*unsere Geschichte*' (meaning '*our history*' in English), within the same latent segment. Overall, Seg2Seg achieves good source-to-target mapping through adaptive aggregation and emission.

**Expectation Training**  Figure 12 shows the mapping during training. In expectation training, Seg2Seg considers all possible latent segments, with the number of segments ranging from $1$ to $J$, shown in Figure 12(a) and 12(b). By incorporating the constraint of the latency loss on the number of segments, Seg2Seg can effectively learn to aggregate an appropriate number of latent segments. Furthermore, Figure 12(c) demonstrates the probability that each target token can pay attention to the source token (referred to as $\mathcal{M}_{ij}$ in Eq.(7)). This expectation aligns well with the inference process depicted in Figure 11(c), thereby highlighting the effectiveness of expectation training.

## B.3 Case Study

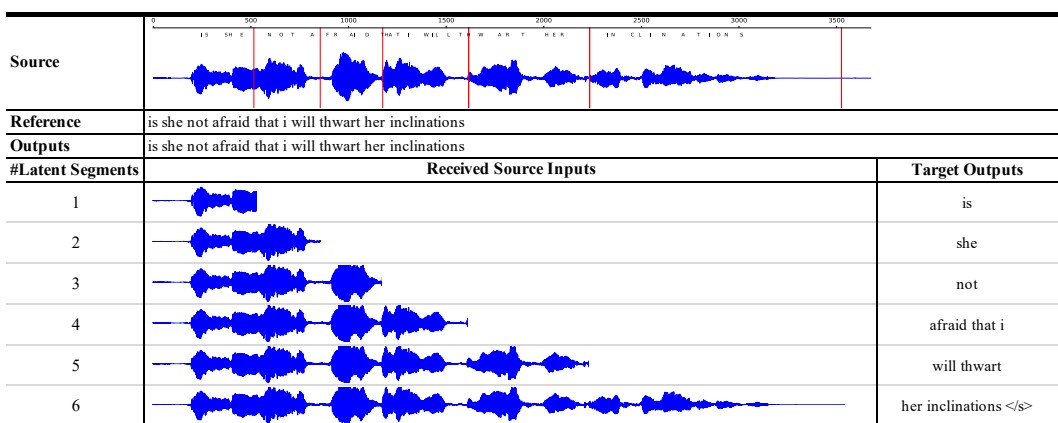

Figure 13: Case study (#4992-23283-0015 in LibriSpeech) of Seg2Seg on streaming ASR task. To demonstrate the simultaneous generation process, we correspond target outputs and received source inputs to show the moments of generating each target token.

| Source | ein  Lot@@  to@@  spieler  aus  Har@@  vey  ist  diesen Monat dra@@ n | |
|---|---|---|
| *English meaning* | *a*    *_Lotto*    *_player*  *_from*    *_Harvey*    *_is*  *_this*    *_month*    *_turn* | |
| **Reference** | a Harvey lotto player is in the month . | |
| **Outputs** | a Lotto player from Harvey is this month . | |
| **#Latent Segments** | **Received Source Inputs** | **Target Outputs** |
| 1 | ein | a |
| 2 | ein  Lot@@  to@@ | Lot@@ to |
| 3 | ein  Lot@@  to@@  spieler | player |
| 4 | ein  Lot@@  to@@  spieler  aus  Har@@ | from |
| 5 | ein  Lot@@  to@@  spieler  aus  Har@@  vey | Har@@ vey |
| 6 | ein  Lot@@  to@@  spieler  aus  Har@@  vey  ist | is |
| 7 | ein  Lot@@  to@@  spieler  aus  Har@@  vey  ist  diesen Monat dra@@ n | this mon@@ th .  |

Figure 14: Case study (#1366 in WMT15 De→En) of Seg2Seg on SimulMT task. To demonstrate the simultaneous generation process, we correspond target outputs and received source inputs to show the moments of generating each target token.

Figure 13, 14 and 15 visualize the simultaneous generation process of Seg2Seg on the cases from streaming ASR, SimulMT and SimulST. In Streaming ASR and SimulST, for a clear illustration, we use an external offline speech-text alignment tool[11] to align the transcription with the speech sequence, and the aligned transcription is displayed above the speech waveform.

---

[11]https://pytorch.org/audio/main/tutorials/forced_alignment_tutorial.html

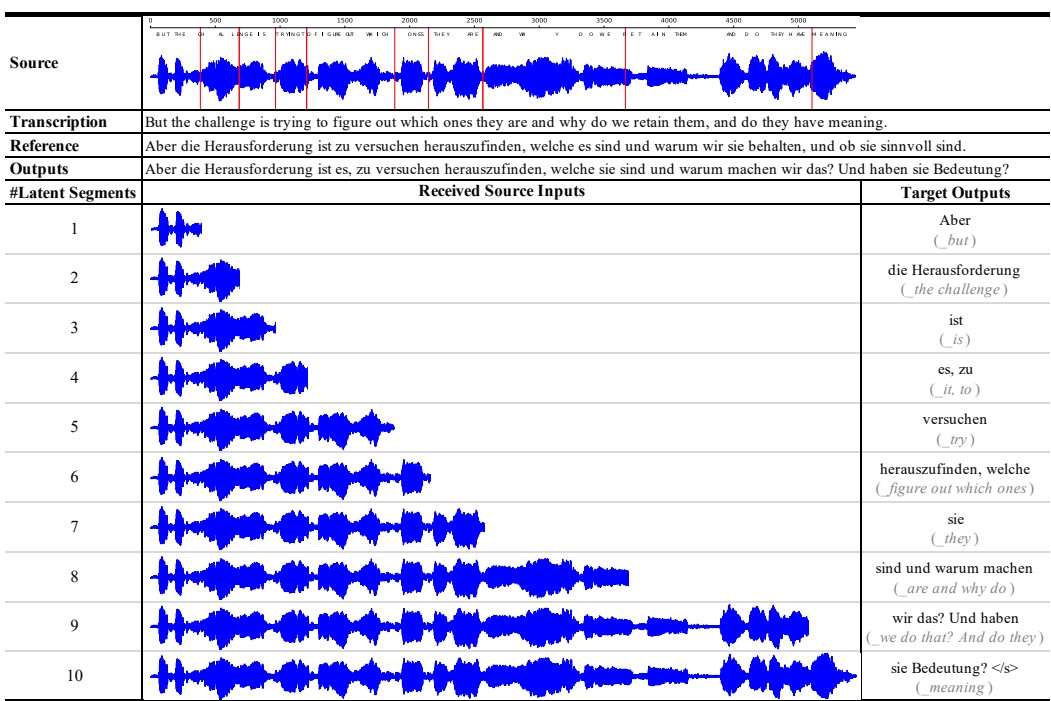

Figure 15: Case study (#1166_119 in MuST-C En→De) of Seg2Seg on SimulST task. To demonstrate the simultaneous generation process, we correspond target outputs and received source inputs to show the moments of generating each target token.

**Case of Streaming ASR**  As shown in Figure 13, the target sequence and source sequence in streaming ASR often exhibit a predominantly monotonic relationship, and Seg2Seg can handle this monotonic mapping well. Seg2Seg can segment and aggregate source sequences at speech boundaries, and then emit the corresponding target sequences accurately.

**Case of SimulMT**  Figure 14 shows a case on SimulMT. Seg2Seg can generate the target token after receiving the corresponding source tokens, e.g., generating '*Lot@@ to*' after receiving '*Lot@@ to*', generating '*player*' after receiving '*spieler*' and generating '*Har@@ vey*' after receiving '*Har@@ vey*', which effectively ensures the generation quality of SimulMT. Besides, for some related target tokens, especially the subwords after the bpe operation, Seg2Seg can emit them together from the same latent segment, thereby achieving lower latency.

**Case of SimulST**  Figure 15 shows a case on SimulST, which is more challenging as the source and target sequences involve different modalities and languages. Despite these evident differences, Seg2Seg demonstrates its capability to find the reasonable generating moments, such as generating '*herauszufinden*' after receiving '*figure out*' in the speech, and generating '*Bedeutung*' after receiving '*meaning*' in the speech. This is mainly attributed to expectation training, which explores all possible mappings in training, allowing Seg2Seg to learn to aggregate and emit at reasonable moments. As seen, Seg2Seg aggregates the related speech frame into the same latent segment and will not break the acoustic integrity of the speech. For emission, Seg2Seg can accurately determine whether a latent segment can emit the target token, where almost all emitted target outputs are correspond to the source speech contained in the latent segment.

# C   Latency Metrics

For the latency evaluation of simultaneous generation task, we use mean alignment delay for streaming ASR and average lagging for SimulMT and SimulST.

**Mean Alignment Delay** [40] is defined as the average word time difference between the ground-truth alignments (speech and transcription) and generating moments:

$$D_{\text{mean}} = \frac{1}{|\mathbf{y}|} \sum_{i=1}^{|\mathbf{y}|} \left( \hat{t}_i - t_i \right), \tag{18}$$

where $t_i$ is the ground-truth alignment of $y_i$, and $\hat{t}_i$ is the generating moment of $y_i$.

**Average Lagging (AL)** [4] evaluates the average number of tokens (for SimulMT) or speech duration (for SimulST) that target outputs lag behind the source inputs. We use $t_i$ to denote the generating moments of $y_i$, and AL is calculated as:

$$\text{AL} = \frac{1}{\tau} \sum_{i=1}^{\tau} t_i - \frac{i-1}{|\mathbf{y}| / |\mathbf{x}|}, \quad \text{where} \ \ \tau = \operatorname*{argmin}_i \left( t_i = |\mathbf{x}| \right). \tag{19}$$

In addition to average lagging, we also use some other latency metrics for SimulMT and SimulST, described as follow.

**Consecutive Wait (CW)** [11] evaluates the average number of source tokens waited between two target tokens, i.e., the number of segments:

$$\text{CW} = \frac{|\mathbf{x}|}{\sum_{i=1}^{|\mathbf{y}|} \mathbb{1}_{t_i - t_{i-1} > 0}}, \tag{20}$$

where $\mathbb{1}_{t_i - t_{i-1} > 0}$ counts the number of $t_i - t_{i-1} > 0$, i.e., the number of segments. It is worth mentioning that the latency loss $\mathcal{L}_{latency}$ in training employs the denominator part of the CW metric, as the numerator is a constant.

**Average Proportion (AP)** [17] evaluates the proportion between the number of received source tokens and the total number of source tokens, calculated as:

$$\text{AP} = \frac{1}{|\mathbf{x}| \, |\mathbf{y}|} \sum_{i=1}^{|\mathbf{y}|} t_i. \tag{21}$$

**Differentiable Average Lagging (DAL)** [5] is a differentiable version of average lagging, calculated as:

$$\text{DAL} = \frac{1}{|\mathbf{y}|} \sum_{i=1}^{|\mathbf{y}|} t_i' - \frac{i-1}{|\mathbf{x}| / |\mathbf{y}|}, \quad \text{where} \ \ t_i' = \begin{cases} t_i & i = 1 \\ \max \left( t_i, t_{i-1}' + \frac{|\mathbf{x}|}{|\mathbf{y}|} \right) & i > 1 \end{cases}. \tag{22}$$

# D   Numerical Results

Table 4, 5 and 6 report the numerical results of Seg2Seg on SimulMT and SimulST, where $\lambda$ a hyperparameter that controls the overall latency of Seg2Seg (refer to Eq.(8)).

Table 4: Numerical results of Seg2Seg on De→En SimulMT.

| | SimulMT on WMT15 De→En | | | | |
|---|---|---|---|---|---|
| $\lambda$ | CW | AP | AL | DAL | BLEU |
| 0.4 | 1.78 | 0.60 | 2.21 | 4.52 | 27.90 |
| 0.3 | 2.30 | 0.63 | 3.78 | 5.83 | 29.28 |
| 0.2 | 2.63 | 0.71 | 5.01 | 7.06 | 30.67 |
| 0.1 | 5.07 | 0.81 | 7.13 | 13.89 | 31.44 |
| 0.05 | 7.20 | 0.85 | 11.23 | 15.32 | 32.20 |

Table 5: Numerical results of Seg2Seg on En→De SimulST.

| λ | CW | AP | AL | DAL | BLEU |
|---|---|---|---|---|---|
| | SimulST on MuST-C En→De | | | | |
| 0.4 | 588.99 | 0.68 | 672.58 | 1444.73 | 14.47 |
| 0.3 | 653.43 | 0.71 | 893.27 | 1533.80 | 16.98 |
| 0.2 | 831.69 | 0.73 | 1068.11 | 1736.00 | 18.44 |
| 0.1 | 1563.63 | 0.79 | 1602.01 | 2477.97 | 21.75 |
| 0.05 | 2532.03 | 0.92 | 3125.51 | 4121.58 | 23.63 |

Table 6: Numerical results of Seg2Seg on En→Es SimulST.

| λ | CW | AP | AL | DAL | BLEU |
|---|---|---|---|---|---|
| | SimulST on MuST-C En→Es | | | | |
| 0.4 | 682.84 | 0.71 | 772.23 | 1645.60 | 20.72 |
| 0.3 | 874.32 | 0.74 | 1129.03 | 1891.32 | 23.96 |
| 0.2 | 1676.52 | 0.80 | 1536.29 | 2724.39 | 26.53 |
| 0.1 | 1963.08 | 0.82 | 2329.22 | 3022.22 | 28.14 |
| 0.05 | 2326.10 | 0.89 | 2838.15 | 4074.75 | 28.69 |

