# OpenReview forum: "Unified Segment-to-Segment Framework for Simultaneous Sequence Generation"
_NeurIPS.cc/2023/Conference — NeurIPS 2023 poster_

### Official Review · Reviewer_ooDH · 2023-07-05

**Soundness:** 3 good
**Presentation:** 2 fair
**Contribution:** 3 good
**Rating:** 5
**Confidence:** 4

**Summary:**

This paper proposes a unified Seg2Seg framework for simultaneous sequence generation tasks. The framework introduces a latent segment as the pivot between the source and target, exploring all potential mappings via the proposed expectation training. With this approach, the authors achieve high-quality generation with low latency across multiple simultaneous generation tasks, including streaming ASR, SimulMT and SimulST.

**Strengths:**

- This paper introduces latent segments as the pivots to explicitly model the mapping between source and target in simultaneous generation tasks.
- The proposed Seg2Seg framework exhibits better performance across various simultaneous generation tasks compared to existing methods.



**Weaknesses:**

- The current version of the proposed framework lacks some details, making it unclear how the prediction of target tokens is learned.
- As the proposed framework leverages wav2vec 2.0 to convert audio into a sequence of discrete hidden states, streaming ASR and SimulST tasks actually could be viewed as special SimulMT tasks. It is necessary to conduct experiments, such as GSiMT and HMT, to verify the performance of previous SOTA SimulMT methods on these tasks.
- There are some confusing experimental results. Figure 4 shows that the performance of the offline model is better than that reported in the HMT paper, despite using the same training dataset and model structure. It is unclear whether the improvement of the proposed method contributes to the better offline model or the Seg2Seg framework compared to HMT.

**Questions:**

1. In experiments, what values of \lambda are being used?
2. Have you experimented with different coefficients for the mapping and latency training loss?
3. The performance difference between the proposed method and HMT is minimal. Have you directly compared the performance of HMT on streaming ASR and SimulST tasks when using wav2vec 2.0 to convert audio into a sequence of discrete hidden states?

---

> ### Author Rebuttal · Authors · 2023-08-06
>
> Thanks for your careful and valuable comments, and we will refine the paper following your suggestions. Following, we will respond to your questions in detail.
>
> &nbsp;
>
> Q1: It is unclear how the prediction of target tokens is learned?
>
> A1: Seg2Seg leverages the Transformer as the backbone and additionally learns aggregation and emission decisions. For predicting target tokens, Seg2Seg follows the same approach as the Transformer and learns this through traditional cross-entropy loss $L_{ce}$ (in Eq.(10)). Note that the mapping and translation are jointly learned via cross-entropy loss $L_{ce}$.
>
> Thank you for your reminder, we will give a more clear explanation in the next version.
>
> &nbsp;
>
> Q2: Streaming ASR and SimulST tasks actually could be viewed as special SimulMT tasks. Have you directly compared the performance of HMT on streaming ASR and SimulST tasks when using wav2vec 2.0 to convert audio into a sequence of discrete hidden states?
>
> A2: Indeed, we have considered the possibility of evaluating the performance of HMT and GSiMT on streaming ASR and SimulST tasks as you pointed out. **However, it is not feasible to directly apply SimulMT methods, such as HMT and GSiMT, to streaming ASR and SimulST via introducing a wav2vec, mainly because SimulST and SimulMT have significant differences in sequence length, and SimulMT methods often involve heuristic design about sequence length.** For streaming ASR and SimulST, although wav2vec2.0 can convert audio into a sequence of discrete hidden states, the sequence length of speech hidden states is still significantly longer the text sequence length in SimulMT. For example, the average source sequence length of WMT15 De-En SimulMT is 20, while the average length of speech hidden states of MuST-C En-De is as high as 300. This difference makes SimulMT methods such as HMT and GSiMT unable to process speech inputs. The reason will be described in detail below.
>
> - For HMT, it involves upsampling the target sequence K times, corresponding to K states, with each state representing a translating moment in the source sequence. Therefore, K is dependent on the length of the source sequence. In SimulMT, HMT requires a maximum upsampling of only 20 times. However, for streaming ASR and SimulST, HMT will require a maximum upsampling of the target sequence by 300 times, with each state corresponding to one translating moment among speech features. The 300 times upsampling is completely unrealistic and unacceptable. Therefore, the HMT framework can only be applied to shorter source sequences in SimulMT and is difficult to directly apply to SimulST.
> - As for GSiMT, its original article already pointed out its high training complexity, requiring cyclic forward the entire model I*J times, where I and J are the lengths of the source and target sequences, respectively. When the source sequence length increases from 20 in SimulMT to 300 in SimulST, the overall training time and resource consumption increase significantly, making GSiMT unviable for SimulST.
>
> Overall, the construction of previous SimulMT methods often involves task-specific heuristics on sequence and thus results in the limitations of applying SimulMT methods to streaming ASR and SimulST tasks via introducing a wav2vec2.0. This is also the primary motivation behind the development of Seg2Seg, that is, to develop a unified simultaneous generation framework that can be applied universally across multiple tasks/sequences. Thank you for your reminder, we will add an explanation of the limitations of applying the SimulMT method to SimulST in the next version.
>
> &nbsp;
>
> Q3: In Figure 4, the performance of the offline model is better than that reported in the HMT paper?
>
> A3: As far as we know, the offline model in the HMT paper uses uni-directional attention (mentioned in the original HMT). The offline model we report uses the standard Transformer, that is, bi-directional attention (in order to be consistent with the SimulST setting), so it is normal to be better than the offline model in HMT paper. Actually, our proposed method and HMT use exactly the same settings, and the HMT performance we implemented is almost the same as the original reported results.
>
> In addition, we additionally compared Seg2Seg and strong baselines on IWSLT15 English-Vietnamese SimulMT, and the results are reported in the PDF uploaded by rebuttal. The results show that Seg2Seg also surpasses previous strong baselines, especially at low latency, demonstrating the advantage of Seg2Seg over SimulMT.
>
> Furthermore, the results in Figure 8 (always be used to evaluate the policy quality) also show that Seg2Seg achieves a more accurate policy than HMT.
>
> Therefore, I hope this experiment can dispel your doubts about whether most of the improvement of Seg2Seg comes from the improvement of translation quality. Thank you for pointing out this issue, we will emphasize the setting of the offline model in the next version to avoid misunderstandings.
>
> &nbsp;
>
> Q4: In experiments, what values of $\lambda$ are being used?
>
> A4: $\lambda$ is introduced in Eq.(9), which is used to constrain the delay of Seg2Seg. Appendix D reports the values of $\lambda$ and the corresponding numerical results in SimulMT and SimulST experiments. For streaming ASR, following previous work, the results in Table 1 are the generation quality at the lowest latency ($\lambda=0.4$). Based on your suggestion, we will add the $\lambda$ setting to the experimental setup.
>
> &nbsp;
>
> Q5: Have you experimented with different coefficients for the mapping and latency training loss?
>
> A5: We did not add any weight coefficients before $L_{ce}$ and $L_{latency}$, and Seg2Seg has already achieved superior performance under this setting. Searching on the coefficients may lead to further improvements, which we leave for future work.
>
> &nbsp;
>
> If our responses answer your questions well and reassure your concerns, we would appreciate if you could reassess our work and increase the rating.

---

> > ### Comment · Reviewer_ooDH · 2023-08-18
> >
> > Thank you for the clarification. I appreciate the author's rebuttal, and it has convinced me to revise my review score. I will now increase my rating from 4 to 5.

---

> > > ### Author Response · Authors · 2023-08-19
> > >
> > > Thanks a lot for your reply and approval.

---

### Official Review · Reviewer_CEsh · 2023-07-07

**Soundness:** 3 good
**Presentation:** 3 good
**Contribution:** 3 good
**Rating:** 6
**Confidence:** 3

**Summary:**

This paper presents an approach (Seg2Seg) for simultaneous sequence generation, which attempts to balance high generation quality with low latency. Rather than processing the entire source sequence at once (incurring high latency), the model splits it into segments. Latent segments are used to aggregate source tokens and generate the corresponding target sub-sequence. While latent segments are discrete for inference, Seg2Seg uses expectation training to learn optimal segment boundaries, combining the cross-entropy loss with a latency loss. Experiments are conducted on streaming automatic speech recognition, simultaneous machine translation and simultaneous speech translation.

**Strengths:**

The model introduced in this paper learns when to stop aggregating information based on latent segments instead of using pre-defined heuristics.

The approach is applicable to multiple simultaneous generation tasks (written and spoken modalities).

Over the 3 considered tasks, the proposed approach generally has favorable BLEU/lagging trade-offs compared to many baselines.

**Weaknesses:**

The notion of adaptive wait policies has been explored in previous work (cited by the authors). This work incrementally improves on known concepts.

Some figures are quite small and might be difficult to read if the paper is printed out.

**Questions:**

How would the approach work between languages with a high amount of reordering (e.g. English (subject-verb-object) and Japanese (subject-object-verb))?

Instead of expectation training, did you consider using the gumbel-max trick? If so, what did you observe?

Why not include [1] (already cited) as a machine translation baseline?

[Minor] In figure 4, why is the Y axis label "BLEU score" for SimulMT and "SacreBLEU" for SimulST?

[1] Arivazhagan et al. Monotonic Infinite Lookback Attention for Simultaneous Machine Translation. ACL. 2019

**Limitations:**

Yes, limitations are addressed.

---

> ### Author Rebuttal · Authors · 2023-08-06
>
> Thanks for your careful and valuable comments, and we will refine the paper following your suggestions. Following, we will respond to your questions in detail.
>
> &nbsp;
>
> Q1: About the notion of adaptive wait policies?
>
> A1: Indeed, I agree with you that the concept of adaptive policies existed before. However, it is important to highlight the key distinction of Seg2Seg. Our focus is that the previous adaptive policies are adaptive on READ/WRITE decisions, but the whole framework (training or some attention design) always involves some task-specific heuristics, such as locality in ASR and similar sequence lengths in MT. These heuristics often limit the direct applicability of previous adaptive policies across multiple tasks.
>
> Therefore, The motivation and innovation of Seg2Seg lie in the development of a universal and adaptive unified framework that can handle various simultaneous tasks, which avoids artificially setting some task-specific heuristics for different tasks and further makes multi-task learning possible.
>
> Thanks for your suggestion, we will emphasize the difference between Seg2Seg and previous adaptive policies in the next version.
>
> &nbsp;
>
> Q2: How would the approach work between languages with a high amount of reordering?
>
> A2: Actually, word reordering is one of the major challenges for both SimulMT and SimulST. To address this issue, Seg2Seg uses the segment as bridge instead of taking the token as the unit like the previous methods, so Seg2Seg will wait until the received words can be aggregated into segments and then translate the target sequence segment-by-segment. This allows Seg2Seg to handle certain levels of local reordering. For instance, in source languages with a word order like 'SVO,' Seg2Seg aggregates the 'VO' into one segment and emits the corresponding 'OV' segment in the target 'SOV' language. For extremely long-distance reordering under low latency requirements, Seg2Seg will generate a target translation with a similar word order to the source sequence, thus balancing translation quality and latency.
>
> In the main experiment, we specially selected the most challenging translation languages for simultaneous generation, German and English, where German is the SOV language and English is the SVO language. Both SimulMT and SimulST results between German and English demonstrate the superior performance of Seg2Seg on languages with such differences in word order.
>
> &nbsp;
>
> Q3: Did you consider using the gumbel-max trick instead of expectation training?
>
> A3: Indeed, the Gumbel-Max trick can facilitate backpropagation in Seg2Seg, enabling the learning of aggregation and emission. While we haven't explored the use of the Gumbel-Max trick in our current work, we sincerely appreciate your valuable proposal. We will certainly consider incorporating the Gumbel-Max trick in our future research endeavors to potentially enhance the performance of our approach. Thank you for bringing this to our attention.
>
> &nbsp;
>
> Q4: Why not include MILk[1] as a machine translation baseline?
>
> A4: When Monotonic Infinite Lookback Attention (MILk) [1] was proposed, it applied monotonic attention on the LSTM architecture. The subsequent Monotonic Multi-head Attention (MMA) [2] is the Transformer implementation of MILk (using the same method), and it also achieves better performance than MILk. Considering that MMA and MILk use the same method, and propose that Seg2Seg is also based on Transformer, we did not add MILk in the main experiment.
>
> Thanks for your suggestion, we will emphasize the relationship between MMA and MILk in the system setting section in the next version.
>
> Reference:
>
> [1] ACL 2019: Monotonic Infinite Lookback Attention for Simultaneous Machine Translation.
>
> [2] ICLR 2020: Monotonic Multihead Attention.
>
> &nbsp;
>
> Q5: Why is the Y axis label "BLEU score" for SimulMT and "SacreBLEU" for SimulST?
>
> A5: Apologies for the confusion caused, and the inconsistency in the label name is a writing typo. Thanks for your observation, we will fix it in the next version.
>
> &nbsp;
>
> Q6: About some figures size?
>
> A6: Thank you for your careful inspection and valuable suggestions. We will enlarge the figures based on your suggestions.
>
> &nbsp;
>
> If our responses answer your questions well and reassure your concerns, we would appreciate if you could reassess our work and increase the rating.

---

> > ### Comment · Reviewer_CEsh · 2023-08-21
> >
> > Thank you for your response. I may consider increasing my rating, although I need to take a closer look at the paper and related work again. Nevertheless, I am still leaning towards acceptance.
> >
> > > we specially selected the most challenging translation languages for simultaneous generation, German and English
> >
> > I am overall satisfied with your answer to Q2, although this statement may be too strong. You can argue it is challenging, but "most challenging" is difficult to prove.

---

> > > ### Author Response · Authors · 2023-08-21
> > >
> > > Thanks a lot for your valuable feedback and for considering a potential increase in the rating for our paper. We sincerely appreciate your time and attention to our work.

---

### Official Review · Reviewer_ZDUt · 2023-07-07

**Soundness:** 3 good
**Presentation:** 4 excellent
**Contribution:** 4 excellent
**Rating:** 8
**Confidence:** 4

**Summary:**

This paper proposes a method for simultaneous sequence to sequence modeling.
The proposed approach introduced a latent segment, which represents the number of consecutive reads followed by consecutive writes.
The method is applied to three simultaneous sequence to sequence modeling task, streaming ASR, simultaneous MT and simultaneous ST.
Results demonstrate better latency-quality tradeoffs compared to a number of baselines. Finally, the framework is applied in a multitask setting (ASR + MT + ST) where it is shown that combining more tasks presents better latency-quality tradeoffs.

**Strengths:**

* the method nicely generalizes previous ideas on simultaneous modeling and is broadly applicable to both ASR and translation
* the main results are strong compared to the state of the art.
* the ablation analyses convincingly show that the improvements come from the proposed approach

**Weaknesses:**

See more detailed comments/questions below on how the paper can be improved.

**Questions:**

* In eq 1, Emb (xj ) is used. This design choice is not justified in the paper. For example, why not use encoder output states, etc.
* the simultaneous MT experiments only concern one language direction. The results would be more convincing with at least two directions.
* the streaming ASR results only have one datapoint. It may be interesting to have a range of latency/quality points (even though it doesn't seem to be common practice in the streaming ASR literature).

Some presentation suggestions:
* 48: "without any heuristics": some of the previous work is also based on models rather than heuristics. Suggest rephrasing to "without any task-specific assumptions"
* 92: "Details are introduced following.": remove or "Details are introduced in the following sections."
* 155: "By jointly trained": "By jointly training"

---

> ### Author Rebuttal · Authors · 2023-08-06
>
> Thanks for your careful and valuable comments, and we will refine the paper following your suggestions. Following, we will respond to your questions in detail.
>
> &nbsp;
>
> Q1: The design choice of Emb(xj) instead of using encoder output states?
>
> A1: Thank you for raising a question regarding the calculation of the aggregation probability in Eq. (1). The decision to use embeddings for predicting the aggregation probability is based on two considerations:
>
> (1) **Efficiency**: In a simultaneous system, we want to make decisions with the least computational cost. If the encoder output is used to predict the aggregation probability, then when each new source token arrives, we need to forward the whole encoder to make a decision, resulting in higher computational overhead. Conversely, by employing embeddings for predicting aggregation probability, we only need to forward the embedding for each new token when making decisions. The complete encoder is forwarded only after deciding to aggregate a latent segment.
>
> (2) **Consistency**: After aggregating into latent segments, we need to calculate the emission probability. This step needs to be completed before the decoder, because the mapping (aggregation and emission probability) needs to be trained jointly with cross-attention. Therefore, the emission probability needs to be predicted via target embedding, i.e., within the embedding space. To maintain consistency, we should predict the aggregation probability and latent segment representation in the embedding space as well, ensuring that Eq.(3) is computed within the embedding space.
>
> Thanks for your reminder, we will provide further justifications for this design choice in the next version.
>
> &nbsp;
>
> Q2: Simultaneous MT experiments on other language direction?
>
> A2: Due to space constraints, we only show the SimulMT results on the most widely used and difficult benchmark WMT15 De-En. We also conducted SimulMT experiments on IWSLT15 English-Vietnamese. The table below shows the performance of Seg2Seg on IWLST15 En-Vi, and the BLEU-AL curves of Seg2Seg and strong baselines are reported in the PDF uploaded in the rebuttal. The results show that Seg2Seg also outperforms the previous state-of-the-art SimulMT method HMT [1].
>
> | AL   | BLEU  |
> | ---- | ----- |
> | 2.76 | 28.10 |
> | 4.01 | 28.75 |
> | 6.54 | 28.82 |
> | 8.65 | 28.88 |
> | 10.2 | 28.90 |
>
> Thanks for your suggestion, we will add the SimulMT results on IWLST15 En-Vi in the next version.
>
> Reference:
>
> [1] ICLR 2023: Hidden Markov Transformer for Simultaneous Machine Translation.
>
> &nbsp;
>
> Q3: About datapoint in streaming ASR results?
>
> A3: Indeed, as you said, the common practice of the streaming ASR literature is to report one datapoint with a minimum latency. This is mainly because the source and target sequences in ASR are monotonic, and higher generation quality can be achieved at lower latency. In Table 1, we report the performance of Seg2Seg at extremely low latency. Considering that Seg2Seg is already close to offline performance at this latency and for fair comparison with previous works, we only report one datapoint. According to your suggestion, we will add more datapoints in the next version.
>
> &nbsp;
>
> Q4: About presentation suggestions?
>
> A4: Thanks for your careful inspection and valuable suggestions. We will refine the paper based on your suggestions.
>
> &nbsp;
>
> Thanks again for your careful and valuable comments, and we will refine the paper following your suggestions.

---

> > ### Comment · Reviewer_ZDUt · 2023-08-20
> > **thanks for the detailed rebuttal + one minor suggestion**
> >
> > Thank you for the detailed answer. All my concerns are addressed.
> > Regarding the additional experiment on En-Vi, can you make sure that the latency ranges are meaningful? It seems to be that beyond 6 AL, everything can be bucketed into "offline". What is more interesting is what happens at lower latency ranges.
> > Note that the En-Vi is quite low resource. To make the experimentation even stronger, you may also want to report results on WMT en-fr as in https://arxiv.org/abs/1906.05218 and compare to that.

---

> > > ### Author Response · Authors · 2023-08-21
> > >
> > > Thanks a lot for your reply!
> > >
> > > For En-Vi, the latency of all methods is between 3-8, which is mainly due to language characteristics (Vietnamese is often longer than English). For example, the extremely low latency wait-1 policy also gets AL=3 latency.
> > >
> > > Thanks again for your suggestion, we will add the SimulMT experiments on En-Fr. While considering the approaching rebuttal deadline, we will include it in the final version.

---

### Official Review · Reviewer_D731 · 2023-07-07

**Soundness:** 3 good
**Presentation:** 3 good
**Contribution:** 2 fair
**Rating:** 6
**Confidence:** 4

**Summary:**

The paper addresses the task of simultaneous sequence generation in real-time scenarios, where the target sequence is generated while receiving the source sequence. The authors propose a unified segment-to-segment framework called Seg2Seg to tackle this challenge. Seg2Seg aims to learn the optimal moments for generating by alternating between waiting for a source segment and generating a target segment, using a latent segment as the bridge between the source and target. The proposed framework employs expectation training to explore all potential source-target mappings and achieve adaptive and unified learning. The authors applied their proposed method to three real-time tasks: Streaming Automatic Speech Recognition (ASR), Simultaneous Machine Translation (SimulMT), and Simultaneous Speech Translation (SimulST). The experimental results demonstrate that their approach achieves a favorable balance between performance and latency in all three tasks.


**Strengths:**

The authors aim to introduce a unified framework for training and inference in real-time tasks, and they have successfully achieved this unified approach.

**Weaknesses:**

1. The proposed approach lacks novelty as it bears similarities to the MMA framework.
2. The efficiency of the proposed method is relatively low as it requires backpropagation through multiple training objectives, resulting in increased training time.


**Questions:**

1. Although the proposed approach shares similarities with the MMA framework, the paper does not explicitly mention the similarities and differences between the two methods. Could the authors provide more insight into how their approach differs from and aligns with the MMA framework?
2. It appears that the training cost of this framework is higher compared to fixed strategies like Wait-k. Could the authors provide a comparison of training time and training cost between their proposed approach and fixed strategies?

---

> ### Author Rebuttal · Authors · 2023-08-06
>
> Thanks for your careful and valuable comments, and we will refine the paper following your suggestions. Following, we will respond to your questions in detail.
>
> &nbsp;
>
> Q1: The difference and superiority of Seg2Seg compared to MMA?
>
> A1: Compared with MMA, there are several fundamental distinctions that make Seg2Seg a more innovative and superior solution:
>
> 1. **Strong generalization across multiple tasks**: MMA was originally introduced in SimulMT, which is good at aligning source and target sequences of similar length in units of tokens. Since its dynamic programming involves token-by-token multiplication operations, it is difficult for MMA to work when the source sequences are very long and is in a different modality than the target sequence. As a result, many additional heuristics are required to be added to the source sequence when MMA is applied to SimulST, e.g., fixed pre-decision of 280ms. On the contrary, Seg2Seg introduces natural segments instead of tokens as a bridge between the source and the target, so it is not affected by the length and modality of the source sequence and thus can be applied to various simultaneous generation tasks such as streaming ASR, SimulMT and SimulST with a unified framework without any heuristic rules. Moreover, this unified framework facilitates multi-task learning, which is more helpful for difficult tasks like SimulST.
>
> 2. **Different decision-making methods**: MMA uses monotonic attention to enable each attention head to independently calculate token alignments, while Seg2Seg takes a different approach by directly determining whether to aggregate latent segment and emit the target token at the embedding level. Many previous works [1,2,3] have shown that the decision-making method of MMA will be affected by some abnormal heads, because MMA can output only when all heads (e.g., 6 layers * 8 heads = 48 heads totally) decide to WRITE, and must wait even only one head select READ. In contrast, Seg2Seg only needs to make a decision once before the encoder and decoder, so as to obtain a more stable decision.
>
> 3. **Less decision-making consumption**: Completely different decision-making methods make Seg2Seg have less decision-making consumption during inference, compared to MMA. For each READ/WRITE decision, MMA needs to forward the entire model, because the READ/WRITE decisions are made collectively by all the heads of the decoder. Denoting the source and target lengths as $J$ and $I$ (corresponding to $J$ READ actions and $I$ WRITE actions), MMA needs forward $(J+I) \times (Enc Embdding+Encoder+Dec Embdding+Decoder) $ in total. However, Seg2Seg has a completely different way of making decisions, deciding whether to aggregate latent segments at the encoder embedding and subsequently deciding whether to emit at the decoder embedding. Therefore, Seg2Seg only needs forward $J \times Enc Embdding + K \times Encoder + I \times Dec Embdding + I * Decoder $ where $K$ is the segment number, thereby reducing decision-making consumption.
>
> 4. **Different latency control**: Seg2Seg and MMA also differ in their approaches to control latency. MMA constrains the average latency through DAL metric. In contrast, Seg2Seg controls the latency more intuitively by learning the segment number, directly influencing the number of READ/WRITE alternations during inference, which better aligns with the user's perception of latency in a simultaneous system [4].
>
> Overall, **Seg2Seg is different from MMA in its unique READ/WRITE decision-making and latency-controlling**. More importantly, Seg2Seg can adapt to a variety of simultaneous tasks without any modification, which sets Seg2Seg apart from MMA. Following your suggestions, we will provide a detailed exposition of the differences between Seg2Seg and MMA in the next version.
>
> Reference:
>
> [1] ICLR 2020: Monotonic Multihead Attention.
>
> [2] ACL 2022: Modeling dual read/write paths for simultaneous machine translation.
>
> [3] Interspeech 2022: Cross-Modal Decision Regularization for Simultaneous Speech Translation.
>
> [4] ACL 2022: Gaussian Multi-head Attention for Simultaneous Machine Translation.
>
> &nbsp;
>
> Q2: Efficiency of the proposed method?
>
> A2: Thank you for your concern about training costs. Indeed, as you pointed out, Seg2Seg, like most adaptive methods, requires quality and latency training objectives to trade off between generation quality and latency, so the training cost of the adaptive method is generally slightly higher than that of the fixed wait-k policy. However, the performance gain brought by the adaptive method compared with the fixed policy is also significant.
>
> To evaluate the training efficiency of Seg2Seg, we show the UPS (updates per second) of Seg2Seg and strong baselines on the SimulMT task in the table below.
>
> | Methods | UPS  |
> | ------- | ---- |
> | Wait-k  | 4.52 |
> | Seg2Seg | 2.61 |
> | HMT     | 1.36 |
> | MMA     | 0.47 |
> | GSiMT   | 0.20 |
>
> Compared with the fixed wait-k policy, Seg2Seg does train slightly slower. However, considering the significant performance improvement brought by Seg2Seg compared with the fixed wait-k, we believe that the slightly higher training cost is acceptable. More importantly, **Seg2Seg is the most efficient in training among previous strong adaptive methods**. The additional training cost of Seg2Seg arises from the calculation of emission probability, while MMA involves looping calculation in each attention head, HMT needs to upsample the target sequence several times, and GSiMT needs to forward the model multiple times in a loop. Therefore, compared with the adaptive methods, Seg2Seg demonstrates a lower training cost while achieving superior performance.
>
> Thanks for your suggestion, we will incorporate a detailed analysis of the training cost in the next version.
>
> &nbsp;
>
> If our responses answer your questions well and reassure your concerns, we would appreciate if you could reassess our work and increase the rating.

---

### Author Rebuttal · Authors · 2023-08-09

Thanks to all the reviewers for their careful and valuable comments.

&nbsp;

Following the reviewers’ suggestion, we additionally report the results on a commonly used SimulMT benchmark IWSLT15 English→Vietnamese (En→Vi), to more comprehensively evaluate the performance of Seg2Seg on SimulMT. As shown in the Figure 1 of the uploaded PDF, Seg2Seg also outperforms strong baselines in IWSLT En→Vi, especially at low latency.

Overall, Seg2Seg achieved better performance than previous strong baselines on the SimulMT results of IWSLT15 En-Vi and WMT15 De-En. At the same time, Seg2Seg is also a unified framework that can be directly used among various simultaneous generation tasks, such as streaming ASR and SimulST, which is another advantage compared to the previous SimulMT methods. Therefore, **superior performance and better versatility on various simultaneous generation tasks are the innovation and value of Seg2Seg**.

&nbsp;

Thanks again to all the reviewers for their valuable comments, and we will refine the paper following your suggestions.

---

### Decision · Program_Chairs · 2023-09-21

**Decision:**

Accept (poster)

**Comment:**

The paper proposes an end-to-end model for simultaneous sequence to sequence generation (e.g. simultaneous speech translation). Compared to previous approaches, it does not rely on task-specific heuristics and enables multi-task learning. The experiments show strong results compared to previous approaches across multiple tasks with better quality vs. latency trade-offs compared to the previous state of the art. While the impact may be somewhat limited due to the specific nature of the task for which the model is proposed, the paper makes a solid contribution to this problem.